# Using Implementation Mapping to develop an intervention program to support veterinarians' adherence to the guideline on *Streptococcus suis* clinical practice in weaned pigs

**Isaura Y. A. Wayop**[1], **Emely de Vet**[2], **Joanne N. Leerlooijer**[3], **Jaap A. Wagenaar**[1,4], **David C. Speksnijder**[1,5]*

1 Faculty of Veterinary Medicine, Department Biomolecular Health Sciences, Division of Infectious Diseases and Immunology, Utrecht University, Utrecht, The Netherlands, 2 Consumption and Healthy Lifestyles Group, Wageningen University and Research, Wageningen, The Netherlands, 3 Knowledge, Technology and Innovation, Wageningen University and Research, Wageningen, The Netherlands, 4 Wageningen Bioveterinary Research, Lelystad, The Netherlands, 5 University Farm Animal Clinic ULP, Harmelen, The Netherlands

* d.c.speksnijder@uu.nl

## Abstract

*Streptococcus suis (S. suis)* infections in weaned pigs are common and responsible for a high consumption of antimicrobials, and their presence is assumed to be multi-factorial. A specific evidence-based veterinary guideline to support the control of *S. suis* in weaned pigs was developed for veterinary practitioners in the Netherlands in 2014. Adherence to the *S. suis* clinical practice guideline helps veterinary practitioners to prevent and control the disease in a systematical approach and thereby improve antimicrobial stewardship and contribute to the prevention of antimicrobial resistance in animals and humans. The impact of such a clinical practice guideline on (animal) disease management depends not only on its content, but also largely on the extent to which practitioners adhere to the clinical guideline in practice. When the *S. suis* guideline was published, no specific activities were undertaken to support veterinarians' uptake and implementation, thereby contributing to suboptimal adherence in clinical practice. As the *S. suis* guideline was comprehensively written by veterinary experts following an evidence-based approach, our aim was not to judge the (scientific) quality of the guideline but to study the possibility to improve the currently low adherence of this guideline in veterinary practice. This paper describes the systematic development, using Implementation Mapping, of a theory-based intervention program to support swine veterinarians' adherence to the *S. suis* guideline. The knowledge, skills, beliefs about capabilities, and beliefs about consequences domains are addressed in the program, which includes seven evidence-based methods (modelling, tailoring, feedback, discussion, persuasive communication, active learning, and self-monitoring) for use in program activities such as a peer-learning meeting and an e-learning module. The intervention program has been developed for practicing swine veterinarians, lasts eight months, and is evaluated through a stepped-wedge design. The Implementation Mapping approach ensured that all

**Data Availability Statement:** All relevant data are within the manuscript and its Supporting information files.

**Funding:** The VET-ENHANCE project (Veterinary guidelines to support antimicrobial stewardship: enhancing implementation through behavioural interventions) was funded by ZonMw (The Netherlands Organisation for Health Research and Development), project number: 541002005.

**Competing interests:** The authors have declared that no competing interests exist.

relevant adopters and implementers were involved, and that outcomes, determinants (influencing factors), and objectives were systematically discussed.

## Introduction

Antimicrobial resistance is a major threat to human and animal health and to the global economy [1,2]. Two major drivers of antimicrobial resistance are the use of antimicrobials selecting for resistant microorganisms and the spread of antimicrobial-resistant microorganisms between humans, animals, and the environment. Reducing the incidence of infectious diseases and optimizing antimicrobial use in both animals and humans are two of the main objectives of the Global Action Plan for Antimicrobial Resistance launched by the World Health Organization in 2015 [3]. Antimicrobial stewardship promotes the appropriate use of antimicrobials to optimize clinical outcomes and control antimicrobial resistance [4].

Following public concerns in the Netherlands in the first decade of this century, various concerted actions were taken at government, farm, and veterinary practice level, and as a result the sale of antimicrobials for use in food-producing animals decreased by approximately 77% between 2009 and 2022 [5,6]. Despite the achieved reduction, further reduction is necessary, in particular on farms with high usage [7,8].

The implementation of clinical practice guidelines is a practical approach that has substantially supported antimicrobial stewardship in human medicine [9]. In 2013, the Royal Dutch Veterinary Association (KNMvD) developed veterinary clinical practice guidelines as part of a voluntary veterinary quality system to support veterinarians in their clinical approach to common animal diseases and prudent antimicrobial use [10]. The veterinary guidelines were published online and introduced to veterinarians through newsletters and other communication channels, but a comprehensive approach to support veterinarians' adoption of these guidelines in clinical practice was lacking. Five of the currently published veterinary guidelines in the Netherlands contain evidence-based recommendations to prevent specific animal diseases in order to decrease antimicrobial use at farm level. One of these veterinary guidelines is the clinical practice *Streptococcus suis (S. suis)* in weaned pigs guideline published in 2014 [11].

*S. suis* infections are seen as one of the major drivers of antimicrobial use in the pig sector, and specifically in weaned pigs. The antimicrobial resistance rates in *S. suis* have increased worldwide since the 1980s [12]. In 2020, 28% of the pig farms in the Netherlands keeping weaned pigs were using more antimicrobials in weaned pigs than the threshold benchmark value for acceptable use as defined by the Netherlands Veterinary Medicines Institute (SDa) [13]. This independent agency was established in 2010 to promote responsible drug use in Dutch animal husbandry in general, and especially antimicrobial usage. The aim of the *S. suis* guideline was to improve antimicrobial stewardship (i.e., responsible use of antimicrobials) in the swine industry, as it includes recommendations about the prevention and (antimicrobial) treatment of *S. suis*. It contains, for example, a comprehensive checklist with risk factors for *S. suis* infections that can be used to prevent *S. suis* outbreaks, although the management of these risk factors will also positively impact the control of other infectious diseases. However, a survey conducted in 2016 showed that the *S. suis* guideline in the Netherlands was used only partly or not at all by most veterinary practitioners surveyed. There has been no comprehensive evaluation of the use of the *S. suis* guideline, and the effect of the guideline on antimicrobial use for the treatment of *S. suis* on farms is unknown [14].

Various factors influence how and to what extent users adopt guidelines: (i) characteristics of the guideline itself (e.g., complexity, procedural clarity); (ii) characteristics of the working

environment/setting/context (e.g., veterinary practice, rules, and regulations); (iii) characteristics of the proposed users (e.g., skills and knowledge), and (iv) support and implementation characteristics (e.g., attitude in the field) [15–18]. To achieve the benefits of clinical guidelines and improve antimicrobial stewardship, it is necessary to promote adherence to guidelines so that they are used sustainably in veterinary practice [19]. Currently, there is no systematic (comprehensive) adoption, implementation, and/or maintenance strategy for the veterinary guidelines in the Netherlands.

In this study, we describe the systematic development of a theory-based intervention program as a final result that aims to support swine veterinarians' adherence to the *S. suis* guideline. As the *S. suis* guideline was comprehensively written by veterinary experts following an evidence-based approach, our aim was not to judge the (scientific) quality of the guideline but to study the possibility to improve the currently low adherence of this guideline in veterinary practice. To this end, we used Implementation Mapping, a systematic step-by-step approach to develop a theory-based intervention program [20]. Implementation Mapping takes Intervention Mapping as a starting point. Intervention Mapping is a widely used framework that guides the design of multi-level health promotion interventions and implementation strategies [21]. It is used to reduce antimicrobial use and characterized as the recipe for antimicrobial stewardship success in human healthcare [22]. The Implementation Mapping framework integrates insights from the implementation science field and provides guidance for analysis of the implementation gap (research-to-practice gap) and how this information can be used to design solutions to address this gap [20,23]. Furthermore, Implementation Mapping has been used to design interventions for comparable (complex) implementation problems in human health, for example the implementation of physical therapy [24] and interventions aimed at reducing overweight and obesity in children and adolescents [25]. Applying Implementation Mapping to improve veterinary clinical practice is novel however.

## Method and results

### Implementation Mapping process

To develop the intervention program, we followed the Implementation Mapping approach consisting of five specific tasks that need to be completed, as shown in Fig 1. Implementation Mapping encourages intervention planners to incorporate three perspectives in planning: 1) a socio-ecological perspective, i.e., considering individuals within their social and physical environment; 2) a participatory perspective, i.e., involving all relevant stakeholders in the planning process; and 3) an evidence and theory perspective, i.e., understanding the problem, developing program objectives, and designing evidence- and theory-based program materials. Box 1 provides an explanation of terminology used in the Implementation Mapping process.

Throughout the process, we worked in a team consisting of professionals from different fields, including practicing veterinarians, specialists from the Dutch Institute for the Rational Use of Medicine, and academic experts in veterinary infectious diseases, qualitative research, human general practice, health communication, and behavior change. To ensure that all program elements (determinants and objectives) were addressed, we circled back to previous tasks if new relevant information was identified. The intervention program took approximately 18 months to develop. In the following sections, we describe how we developed it.

### Task 1: Conduct an implementation needs assessment

The first task of our Implementation Mapping was to conduct a needs assessment to identify barriers to, and facilitators of, successful implementation. In this task, all relevant stakeholders, including program adopters and implementers, were identified and involved.

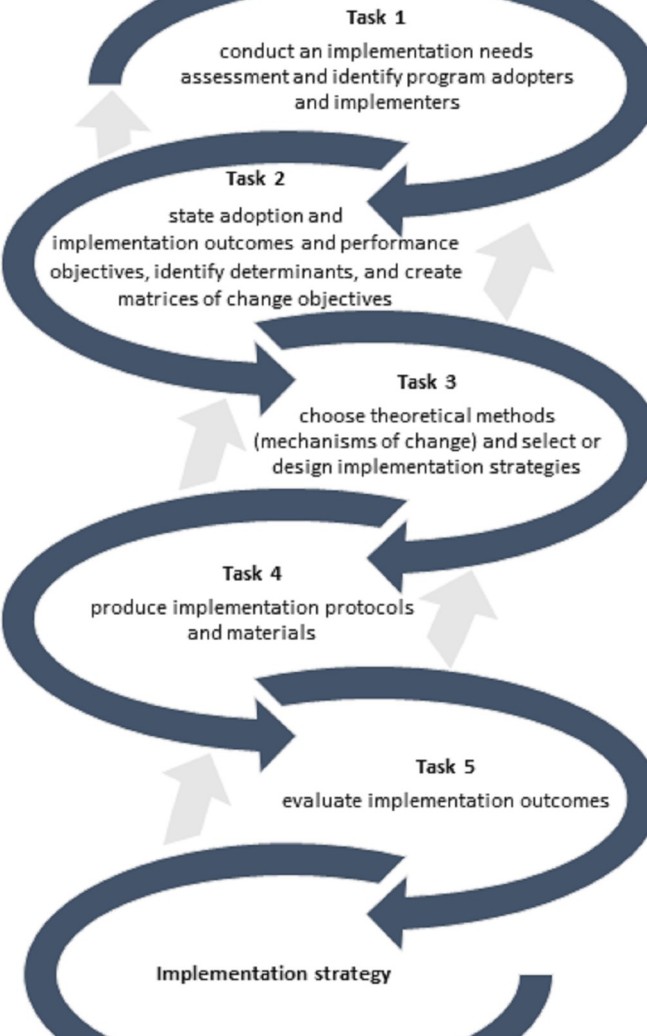

**Fig 1. Implementation Mapping process.** The intervention planners can circle back to previous tasks to ensure that all elements (adopters and implementers, outcomes, determinants, and objectives) are addressed [20].

We started by reviewing the literature on the implementation of clinical guidelines and the planning of health education programs, which have relevance for the *S. suis* guideline. No relevant studies were found about the implementation of veterinary guidelines.

Then, relevant stakeholders were invited for a dialogue with the aim of eliciting their knowledge about the veterinary guidelines, their views on implementation, and other relevant experiences with the veterinary guidelines. We had dialogues with the developers of the *S. suis* guideline (veterinary practitioners and experts in *S. suis* infections), two representatives of veterinary professional associations in the Netherlands, two diplomates of the European College of Porcine Health Management (working at the Faculty of Veterinary Medicine at Utrecht University), a representative of the Netherlands Veterinary Medicines Institute, a

## Box 1. Implementation Mapping Glossary.

### Implementation Mapping glossary

Program adopters: the person(s) taking the decision to use the guideline. For example, in the case of the *S. suis* guideline, a practicing swine veterinarian who has adopted/uses the *S. suis* guideline.

Program implementers: the person(s) applying the guideline long enough in the right way to allow evaluation of whether the guideline meets the perceived needs. For example, in the case of the *S. suis* guideline, a practicing swine veterinarian who has participated in the *S. suis* guideline intervention program.

Program maintenance: the extent to which the program is continued and then becomes part of normal practices and policies.

Needs assessment: a systematic study of discrepancies between what is and what should be in a group and situation of interest.

Performance objectives: the steps, or sub-behaviors, that adopters and implementers must perform to meet the overall adoption and implementation outcomes. They make clear "who has to do what" for the program to be adopted, implemented, and maintained.

Determinant of behavior: an influencing or determining element or modifiable factor that influences the behavior of adopters and implementers.

Domains: determinants grouped in overarching behavioral determinants.

Change objectives: the discrete changes required in each relevant determinant of behavior that will influence achievement of the performance objective. They make clear "what needs to change in the determinant" to achieve the performance objective.

Theory- and evidence-based change method: a general technique or mechanism for influencing the determinants of behaviors and environmental conditions.

representative of the Stichting Geborgde Dierenarts (independent foundation for certifying the quality of veterinarians' veterinary services in the Netherlands), and the Royal GD in Deventer (Dutch Animal Health Services).

The dialogues showed that practicing swine veterinarians were the adopters of the *S. suis* guideline. Farmers and other stakeholders (e.g., nutritionists, inspection authorities) were the beneficiaries of the guideline; they can influence *S. suis* guideline adherence, but they are also influenced by the extent to which veterinarians adhere to the guideline. For example, a farmer whose herd has health issues and is not comfortable with treating the piglets individually (with injections) and therefore prefers group treatments (through food or water) can influence the veterinarian's adherence to the *S. suis* guideline, as this knowledge can influence (the effect of) her/his advice.

**Table 1. Example results qualitative study.**

| Domain | Constructs | Example quote interviews |
|---|---|---|
| **Knowledge** | The veterinarian's knowledge regarding the *S. suis* guideline and handling *S. suis* problems. | "*Then I have to read it again, I cannot remember the guideline clearly.*" |
| **Skills** | The veterinarian's skills/competence/ability regarding implementing the *S. suis* guideline. | "*Often we have the right skills to solve the problem, but sometimes it takes time, but we have those skills to treat the animals but also [educate] the farmer.*" |
| **Beliefs about capabilities** | The veterinarian's self-efficacy regarding implementing the *S. suis* guideline, including self-confidence/professional confidence, self-esteem, and optimism/pessimism. | "*I am not able to solve* S. suis *problems structurally on the farms.*" |
| **Beliefs about consequences** | The veterinarian's anticipated outcomes, consequences, attitudes, and rewards regarding the *S. suis* guideline. | "*The guidelines contain a lot of words, but for a practitioner, for* S. suis *problem farms*[a], *the* S. suis *guideline does not bring me further.*" |
| **Social influences** | The veterinarian's social support and group norms regarding the *S. suis* guideline. These include the opinions and behaviors of colleagues and the farmer. | "*I think colleagues, a big team and peer-consultations, they are more valuable than any guideline.*" |
| **Environmental context and resources** | The veterinarian's environmental constraints and resources/material resources (availability and management) regarding the *S. suis* guideline. These include the veterinary practice's policy, the farm layout, laws and regulations, other stakeholders and advisors, etc. | "*The rules are unclear, there are different laws that are not all consistent, different guidelines in countries, there is a lot, but they all say something different.*" |

The full qualitative study was published separately [26].

[a]In this study, we defined an *S. suis* problem farm as stated in the *S. suis* guideline (use of second choice antimicrobials and/or antimicrobial use above the nationally defined threshold value to treat/control *S. suis* infections) [11]. In the Netherlands, veterinary antimicrobials are classified as first, second, and third choice antimicrobials, where first choice antimicrobials can be prescribed empirically, second choice antimicrobials can be prescribed if it is well reasoned and documented, and third choice antimicrobials can be prescribed to individual animals only after susceptibility testing because of their importance for public health [27].

We decided to develop the intervention for practicing swine veterinarians in the Netherlands, starting from the idea that, through veterinarians' behavior change, ultimately farmers would also benefit from improved animal health and safeguard human health. The dialogues also suggested that it was unknown whether, how, and why the veterinary guidelines were used in practice and how this could be measured. This suggested that more research on practicing swine veterinarians' adoption and usage of the *S. suis* guideline was necessary.

Next, we conducted a qualitative study to get a better understanding of the perceived barriers to, and facilitators of, veterinarians' adherence to the *S. suis* guideline [26]. We interviewed 13 practicing swine veterinarians and five swine farmers and used the theoretical domains framework to explore determinants of veterinarians' decision-making process regarding adherence to, and implementation of, the *S. suis* guideline [17]. In summary, the study results showed that 11 domains of behavioral determinants of guideline adherence were particularly relevant, and together they formed the results of Task 1. Six domains were mentioned consistently in all interviews with veterinarians (knowledge, skills, beliefs about capabilities, beliefs about consequences, social influences, and environmental context and resources), whereas the other domains (motivation and goals, memory, attention and decision processes, nature of the behaviors, social/professional role and identity, and emotion) were mentioned more incidentally during the interviews. This suggests that the six consistent domains are relatively more important for *S. suis* guideline adherence. Table 1 provides an explanation of the six domains.

## Task 2: Identify adoption and implementation outcomes, performance objectives, determinants, and change objectives

In the second task, we identified one adoption and implementation outcome (desired result), performance objectives, and behavioral determinants. We also developed matrices of change

**Table 2. Final performance objectives of the intervention program to improve veterinarians' adherence to the *S. suis* guideline.**

| | Performance objectives intervention *S. suis* guideline for swine veterinarians |
|---|---|
| 1 | The veterinarian writes complete reports and/or administers at least the following information regarding every *S. suis* problem farm:<br>a. the findings from clinical examination and an estimation of the number of affected animals;<br>b. the (probable) diagnosis and the potential results of diagnostics;<br>c. the vaccination status;<br>d. the advice(s) and/or treatment plan;<br>e. motivation for deviation from first choice antimicrobials and the therapy evaluation;<br>f. the number of animals that will be treated and the pens and section(s) in which the animals to be treated are located. |
| 2 | The veterinarian advises the farmer to administer corticosteroids to piglets with nervous disorders caused by *S. suis*. |
| 3 | The veterinarian advises the removal of sick piglets with a probable diagnosis of meningitis caused by *S. suis* from the flock and the provision of water to the piglets. |
| 4 | In principle, the veterinarian prescribes first choice antimicrobials for the treatment of *S. suis*. |
| 5 | The veterinarian prescribes group treatments when 5% or more piglets in the herd/group are affected within 5 days or 4% or more piglets are affected within 24 hours. |
| 6 | The veterinarian identifies an *S. suis* problem farm when (i) the antimicrobial use is above the threshold in weaned piglets and/or (ii) second choice antimicrobials are used to treat meningitis caused by *S. suis*. |
| 7 | The veterinarian advises post-mortem examinations at a first *S. suis* outbreak (at least 2 piglets twice a year, including bacteriological culturing and susceptibility testing). The veterinarian advises structural post-mortem examination (4 times a year, 2 piglets including susceptibility testing) at an existing *S. suis* problem farm. |
| 8 | If autogenous[b] vaccines are used, the veterinarian recommends that piglets are regularly examined for serotyping of *S. suis* by a laboratory. |
| 9 | The veterinarian advises euthanasia of (i) piglets with severe brain symptoms and (ii) piglets that deteriorate within 8 hours or do not recover sufficiently within 48 hours. |
| 10 | The veterinarian uses the recommendations of the *S. suis* guideline for treatment of piglets with meningitis and not for piglets with arthritis. |
| 11 | The veterinarian actively searches for *S. suis* risk factors at the farm (if unknown), for which the checklist in the *S. suis* guideline can be used. On the basis of the identified risk factors, the veterinarian gives tailored advice to prevent *S. suis* problems. |
| 12 | The veterinarian does not use the *S. suis* guideline for weaned piglets as a checklist to comply with inspectors' sanction system, but as a helpful tool in advising swine farmers. |
| 13 | The veterinarian ensures that his/her knowledge about *S. suis* and the rules regarding it is kept up to date by following regular refresher courses or reading scientific literature on this subject. The veterinarian visibly applies this knowledge in his/her advice. |

[b] Inactivated or non-inactivated immunological veterinary medicinal products manufactured from pathogens and antigens obtained from an animal or animals from a farm and used for the treatment of that animal or the animals of that farm in the same locality [28].

objectives, in which performance objectives are crossed with behavioral determinants to develop change objectives.

In our study, the adoption and implementation outcome of the program was defined as practicing swine veterinarians' adherence to the *S. suis* guideline. The results from Task 1 helped us to identify performance objectives, which were discussed in separate sessions with the project team. Reviewing the performance objectives, we also considered their compatibility with practical conditions such as the time period for the intervention program and financial and material resources. This process was completed repeatedly until all the results from Task 1 were incorporated, and finally 13 performance objectives were identified for the intervention (Table 2).

We selected the determinants from four domains (knowledge, skills, beliefs about capabilities, and beliefs about consequences) as targets for our intervention program because of their

**Table 3. Matrix with example of performance objective 6 linked to the change objectives.**

| Performance objective | Knowledge (K) | Skills (S) | Beliefs about capabilities (B) | Beliefs about consequences (C) |
|---|---|---|---|---|
| The veterinarian | The veterinarian . . . | | | |
| **identifies an *S. suis* problem farm (i) when the antimicrobial use is above the defined threshold value in the weaned piglets and/or (ii) when 2<sup>nd</sup> choice antimicrobials are used for meningitis resulting from *S. suis*** | Can list the *S. suis* problem farms in her/his veterinary practice P.6.K.1 | Can show the skills to convince and educate a farmer about having an *S. suis* problem P.6.S.1 | Is convinced that (s)he has the capability to help a farm to achieve good results without the use of 2<sup>nd</sup> choice antimicrobials P.6.B.1. | Is convinced that a *S. suis* problem farm already has a problem before resistance against antimicrobials exists and when a 2<sup>nd</sup> choice antimicrobial is needed P.6.C.1. |
| | Can state the definition of an *S. suis* problem in the *S. suis* guideline P.6.K.2. | | Is convinced that (s)he is capable of convincing and educating a farmer P.6.B.2. | Is convinced that a 2<sup>nd</sup> choice antimicrobial is not necessary for a farm to be successful P.6.C.2. |
| | Can explain that the Defined Daily Dose Animal above 20 is the action value following the Netherlands Veterinary Medicines Institute P.6.K.3 | | | |

changeability (deemed likely that our intervention will influence change in these determinants) and high importance or relevance (they are related to each individual performance objective and were mentioned in all interviews [26]). We cross-linked the performance objectives with the determinants and created the matrix of change objectives. An example of one performance objective and the corresponding change objectives from the theory-informed blueprint is shown in Table 3. The other matrices of the performance objectives linked to the change objectives can be found in S1 Appendix in S1 File.

## Task 3: Select theoretical methods and design implementation strategies

In the third task, the project team selected theory- and evidence-based change methods to achieve the change objectives. A theory- and evidence-based change method (mechanism of change) is a general technique or mechanism for influencing changes in determinants of swine veterinarians' behaviors [21]. To get from the matrices with change objectives (the results of Task 2) to the selection of methods, we grouped the change objectives for each determinant and matched relevant methods to the determinants. Subsequently, the program activities and materials were chosen and, where needed, modified for each corresponding method. Together, they formed the intervention program that fitted with the target population, culture, and context.

Seven theory- and evidence-based change methods were selected that together addressed all change objectives: modelling [29], tailoring [30], feedback [29], discussion [31], persuasive communication [31], active learning [32], and self-monitoring [33]. Most methods addressed multiple change objectives. We coded every change objective, so it was easy to see to which performance objective and determinant they were linked. Table 4 shows the methods, a description, and an example of the change objectives linked to one performance objective, including the codes used to follow the change objectives during the development process.

The final outcome of Task 3 was the design of the intervention program, which consisted of three peer-learning meetings and an individual e-learning module. Peer learning is a group activity in which expert professionals review one another's work, actively give and receive feedback in a constructive manner, teach and learn from one another, and mutually commit to improving performance as individuals, as a team, and as a system.

We chose to incorporate peer learning for two reasons. Firstly, the results of our qualitative study showed that the veterinarians valued the opinion of their peers over the opinion of

**Table 4. Example of change objectives linked to methods in the process to develop the intervention program for the *S. suis* guideline.**

| Method | Description method | Change objectives linked to performance objective 6: the veterinarian identifies an *S. suis* problem farm |
|---|---|---|
| | | The veterinarian . . . |
| **Modelling** | Providing an appropriate role model suitable for practicing swine veterinarians. | Is convinced that (s)he has the capability to help a farm to achieve good results without the use of 2nd choice antimicrobials (B) P.6.B.1. |
| **Tailoring** | Matching components of the intervention to previously measured characteristics of the veterinarians (group) from previous group meetings. | Is convinced that (s)he is capable of convincing and educating a farmer (B) P.6.B.2. |
| **Self-monitoring of behavior** | Prompting the veterinarian to keep a record of specific behaviors. | Can list the *S. suis* problem farms in her/his veterinary practice (K) P.6.K.1 |
| **Feedback** | Giving information regarding the individual veterinarian's adherence to the *S. suis* guideline. | Can state the definition of an *S. suis* problem in the *S. suis* guideline (K) P.6. K.2. |
| **Discussion** | Encouraging consideration of a topic in an open debate. | Is convinced that an *S. suis* problem farm already has a problem before resistance against antimicrobials exists and when a 2nd choice antimicrobial is needed (C) P.6.C.1. |
| **Active learning** | Encouraging learning from goal-driven and activity-based experiences. | Can explain that the Defined Daily Dose Animal above 20 is the action value following the Netherlands Veterinary Medicines Institute (K) P.6.K.3. |
| **Persuasive communication** | Guiding veterinarians toward the adoption of an idea, attitude, or actions by using arguments for example with evidence-based literature. | Is convinced that a 2nd choice antimicrobial is not necessary for a farm (commercial, animal health, animal welfare, mortality, development of antimicrobial resistance, pressure from clients, society, and government) (C) P.6.C.2. |

In this example, we have assigned one method to each change objective, but in practice a change objective can be addressed with multiple methods. The domains and the codes (used to follow the change objectives easily) are mentioned after every change objective. The mentioned domains are knowledge (K), skills (S), beliefs about capabilities (B), beliefs about consequences (C).

scientists [26]. Secondly, peer-learning groups are well evaluated and commonly used by general practitioners in the Netherlands [34] and are also appreciated by veterinarians [35]. The peer-learning meetings were chaired by an independent facilitator (from the Dutch Institute for the Rational Use of Medicine [36]). Besides the facilitator and the participants, the project leader was present to answer questions.

We chose to incorporate an e-learning module for two reasons. Firstly, the results of our qualitative study showed that the veterinarians lacked knowledge about the content of the *S. suis* guideline, which we could offer through e-learning without fixed time and distance limitations (no need to travel) for the participants. Secondly, the use of e-learning modules in human medicine has been shown to be effective in optimizing antimicrobial use [37].

The peer-learning meetings and the e-learning module consist of multiple program activities. One of the incorporated program activities is performance indicators (also known as quality indicators, numeric indicators, quality measures, or key figures), which are measurable items referring to structures, processes, and outcomes of veterinary care regarding the *S. suis* guideline. The performance indicators are used for the participants to self-monitor their behavior and to compare and discuss issues with peers. Table 5 shows more examples of the chosen program activities.

## Task 4: Produce implementation protocols and materials

The fourth task of Implementation Mapping is the development, pre-test, refinement, and production of the materials and protocols needed for the intervention program. Following the Implementation Mapping tasks clarifies what specific messages and materials are needed.

In our study, the protocols and materials were produced by the project leader and external material developers from the Dutch Institute for Rational Use of Medicine, with experience in

**Table 5. Examples of the chosen program activities linked to the change objectives and the method in the different parts of the intervention program.**

| Part intervention program | Example program activity | Example linked change objective | Method |
|---|---|---|---|
| | | The veterinarian. . . | |
| **Meeting 1: start-up** | Movie of role model (practicing swine veterinarian) who encourages farmer to search for *S. suis* risk factors. | Can convince farmer and employees to find *S. suis* risk factors (S) P.11.S.1. | Modelling. |
| | The facilitator asked questions at the start and tailored the meeting if necessary. For example, if a participant did not feel that s(he) had the ability to convince a farmer, (s)he made extra time for this topic and how to approach this problem. | Is convinced that (s)he can influence the farmer about which antimicrobial to use (B) P.4.B.1. | Tailoring. |
| **Meeting 2: self-evaluation** | Participants are asked for performance indicators for their own *S. suis* problem farms. | Can list the *S. suis* problem farms in her veterinary practice (K) P.6.K.1. | Self-monitoring of behavior. |
| **Meeting 3: agreements** | Performance indicators of farmers' own *S. suis* problem farms are shown from previous measurement period till now compared with colleagues. | Is convinced a second choice antimicrobial is not always necessary for a farm to receive good economic results (C) P.4.C.1. | Feedback. |
| | Discussion on farmers' own *S. suis* problem farm. | Can recall the statements about post-mortem examination in the *S. suis* guideline (K) P.7.K.1. | Discussion. |
| **E-learning** | Exercise in e-learning module that encourages searching for answers about the effectiveness of corticosteroids. | Can explain why corticosteroids have a positive effect on the piglet's recovery (K) P.2.K.1. | Active learning. |
| | Case discussion in meeting questions what needs to be in report to the farmer. After discussion, the right comprehensive answers are provided. | Is convinced a good report for the farmer has a value for the farmer (C) P.1.C.1. | Persuasive communication. |

The linked determinants are shown in the change objectives column: Knowledge (K), skills (S), beliefs about capabilities (B), and beliefs about consequences (C). The most important activities in peer-learning meeting 1 are movies, case discussions, and group discussions. The most important activities in peer-learning meeting 2 are a quiz, group-discussions, performance indicators about farmers' own *S. suis* problem farms, and take-home activities. The most important activities in peer-learning meeting 3 are the evaluation of the performance indicators process in terms of farmers' own *S. suis* problem farms and by colleagues, group discussions, and peer feedback. The e-learning takes place between peer-learning meetings 1 and 2. The most important program activities in the e-learning module are active exercises with links to evidence-based literature, cases, tips, cartoons, and multiple quizzes.

producing materials for intervention programs in human medicine. The material developers were well informed about the results of Tasks 1 to 3, introduced in the pig sector, and met regularly until all the following materials were completed.

Peer-learning meeting 1 is the start-up and aims primarily to build trust between the participating veterinarians and between the participants, the facilitator, and the intervention developers. For this first meeting, we developed a presentation about the background of the project, an introduction movie with statements from stakeholders, a manual for the facilitator for the case discussion, and a movie from a practicing swine veterinarian.

Peer-learning meeting 2 is about self-evaluation and aims to discuss performance indicators of *S. suis* problem farms with the respondents and to develop group agreements that could be evaluated in the next meeting. For this second meeting, we developed a quiz, performance indicators, a manual for the participants about the performance indicators, and a manual for the facilitator that included examples of outcomes from the performance indicators, how the results of the performance indicators could be presented, and examples of group agreements.

Peer-learning meeting 3 is the last one and aims to show and evaluate the participants' results and make final individual and/or group agreements. For this third meeting, we developed a manual for the facilitator that included examples of outcomes from the performance indicators over time, how the results of the performance indicators over time could be presented, and examples of individual and group agreements.

The e-learning module is an individual activity that aims to educate the participants about *S. suis*, the *S. suis* guideline, and veterinary guidelines. For the e-learning, we developed a

movie, active exercises with links to evidence-based literature, cases, tips, cartoons, and multiple quizzes.

We used the selected methods for every material that we developed. For example, the target group can identify with the model (parameter of method modelling [21]) represented in the movie: a practicing swine veterinarian, dressed in his normal working clothes (overall and boots), was filmed in a pig pen.

The e-learning module and all the manuals were evaluated and tested by practicing swine veterinarians, a behavioral science specialist, a specialist in *S. suis*, and a specialist in antimicrobial resistance. Changes were made during the process mostly because of practical issues that arose, for example how data from participants and their farmers could be received, and to ensure that the intervention program fitted participants' selected time investments. The total time taken to design and produce the e-learning materials was nine months.

The final result of Task 4 was all the materials developed to prepare and follow the three peer-learning meetings and the individual e-learning module. Table 6 shows the complete content of the peer-learning meetings including the program activities and a short description. Two screen shots from the e-learning module and the quiz questions can be found in S2 Appendix in S1 File and S3 Appendix in S1 File.

## Task 5: Evaluate implementation outcomes

In the fifth and final Implementation Mapping task, the evaluation plan for the intervention is developed. Following the Implementation Mapping tasks clarifies the expected implementation results from the matrix of change objectives. It is recommended to perform a process evaluation (measuring whether the intervention program has been correctly implemented) and an outcome evaluation (measuring the extent to which the intervention program has reached its goals), and to use mixed methods approaches (intentional use of a variety of methods) [38].

In our study, we wanted to evaluate the extent to which the intervention program contributed to veterinarians' intended adherence to the *S. suis* guideline and whether improved guideline adherence translates to better antimicrobial stewardship.

**Process indicators.** We chose to evaluate the intervention program process by feedback spoken directly to the facilitator after every meeting, records of every meeting by the project leader, two questions in the questionnaire, and directly spoken feedback about the e-learning module.

**Outcome indicators.** The performance indicators measure veterinarians' adherence to the *S. suis* guideline and can therefore be used as numeric outcome indicators. These five performance indicators are: antimicrobial use, the ratio of first to second or third choice antimicrobials, the argumentation for second choice antimicrobials, the use of corticosteroids, and the bacterial examination of piglets. The performance indicators have to be measured for two years, divided into three measurement periods. Period 0 encompasses the 12 months before the intervention program starts (the baseline measurement). Period 1 runs from the start of the intervention program and lasts six months; and period 2 begins two months before the last meeting and ends four months after the intervention program has ended.

To measure behavior change, we developed a questionnaire consisting of 63 questions. The questions assess veterinarians' adherence to the 13 performance objectives. This questionnaire has to be filled in before the start of the intervention program and after the last meeting, which is held eight months after the start of the intervention program.

**Trial.** We chose a stepped-wedge cluster trial design [39] for logistical reasons, to prevent simultaneous treatments by the practicing swine veterinarians, and to measure a baseline without having a control group. Following expert opinions of intervention supervisors in human

**Table 6. The content and description of the peer-learning meetings of the intervention program.**

| Meeting | Content | Description |
|---|---|---|
| **Peer-learning meeting 1: start-up** | Introduction | Introduction and first question round. |
| | Introduction movie | Statements and opinions stakeholders and reaction from participants. |
| | Background project | Presentation with introduction of project and researchers. |
| | Movie role model | Movie from practicing swine veterinarian who encourages farmer to search for *S. suis* risk factors. |
| | Case study | Exercise with question about approach *S. suis* problem farm. First in small groups, next discussion in whole group. |
| | Explanation data | Explanation about performance indicators and how to collect the data. |
| | Upcoming meetings | Explanation and schedule upcoming meetings. |
| | Questions | Question round. |
| | Evaluation and closing | Evaluation meeting 1. |
| **Peer-learning meeting 2: self-evaluation** | Introduction | Last updates from facilitator and participants. |
| | Evaluation e-learning | First reactions about e-learning. |
| | Knowledge quiz | Quiz about content *S. suis* guideline. |
| | Explanation performance indicators | Examples of results performance indicators to set expectations. |
| | Results performance indicators | Anonymous data of own *S. suis* problem farms measured in performance indicators are showed. Participants know which data is theirs and can see their results compared to other participants in the group. |
| | Discussion | First reactions and discussion based at the results from the performance indicators. |
| | Group objectives | Group objectives and agreements are discussed. |
| | Questions | Question round. |
| | Evaluation and closing | Evaluation meeting 2. |
| **Peer-learning meeting 3: agreements** | Introduction | Last updates from facilitator and participants. |
| | Recap | Presentation with statements and objectives from participants of previous meetings. |
| | Results own performance indicators | Anonymous data of own *S. suis* problem farms measured in performance indicators and how they are changed in the last six months are showed. Participants know which data is theirs and can see their results compared to other participants in the group. The last slide shows an overview of all the results. |
| | Preparation farm colleague | In small groups (2–3 participants) they discuss each other's biggest *S. suis* problem farms and prepare a presentation to the whole group. |
| | Presentation farm colleague | Every participant presents a *S. suis* problem farm of one of his colleagues to the whole group. |
| | Questions | Question round. |
| | Evaluation and closing | Evaluation meeting 3 and continuation project update. |

medicine, we decided on a minimum of six and a maximum of 12 participants in each peer group. Using a stepped-wedge trial design means that each group starts two months apart, so there is enough time to measure the performance indicators between the meetings. Fig 2 gives an overview of intervention program's timeline, including the three measure periods for the performance indicators and the questionnaires.

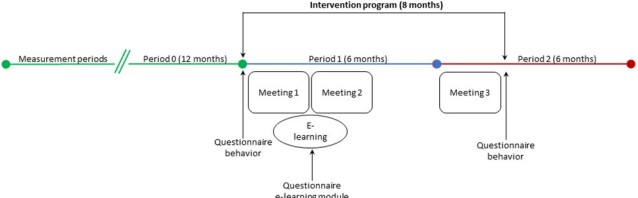

**Fig 2. Overview measurement periods intervention program *S. suis* guideline.** The intervention program consists of three group meetings (of 1.5 hours) and an individual e-learning module (of 2–2.5 hours) and takes eight months to complete. Period 0 is the baseline and lasts 12 months. Period 1 runs from the start of the intervention program and lasts six months. Period 2 covers two months of the intervention program and two months after the intervention program and lasts six months.

## Discussion

Veterinary guidelines, similar to medical guidelines, are not adopted spontaneously by veterinarians when these are made available to them, and the *S. suis* guideline is exemplary of suboptimal adherence. We used the Implementation Mapping approach to systematically develop a theory-based intervention program targeting practicing swine veterinarians in the Netherlands. The aim of the program was to support swine veterinarians' adherence to the *S. suis* guideline in order to improve the uptake of antimicrobial stewardship principles. The program included several activities (e.g., peer-learning, e-learning module) based on seven theoretical behavior change methods to achieve the 13 performance objectives and the 127 change objectives identified by following the Implementation Mapping steps.

Implementation Mapping can be used for both existing clinical practice guidelines [40] and newly developed clinical practice guidelines in the future [21] and proved to be a useful model in our intervention, for two reasons. Firstly, the approach encouraged us as intervention planners to design, run, and evaluate the five tasks in a systematic and structured way and make informed decisions based on evidence and theory [20]. It is not uncommon for intervention developers to focus immediately on modes and methods for intervention delivery (e.g., focusing on developing an app or a website), thus running the risk of using ineffective methods, addressing irrelevant behavioral determinants, or addressing behaviors that are not the most pertinent to the problem at hand. Through Implementation Mapping, we were able to develop a logic model of change where different parts of the intervention were hypothesized to create change in behavioral determinants that would in turn contribute to behavior change in accordance with antimicrobial stewardship principles. Second, in line with Implementation Mapping recommendations [21], practicing swine veterinarians (the adopters) were involved in all stages of the development process to ensure that the intervention was tailored to their needs and preferences. This resulted in an emphasis on peer learning in the program. Veterinarians clearly valued information from their colleagues over information from scientists presented in evidence-based literature or lectures. In human medicine, peer-learning meetings (also called quality circles) are already a major part of continuing professional development and quality improvement [41]. However, there are considerable variations in the effectiveness of peer-learning meetings, in which numerous factors play a role [42]. Factors that contribute to their effectiveness include, but are not limited to, active participants, high quality educational materials, experts' local knowledge, a safe learning environment, and audit and feedback methods [43,44]; we adjusted all of these elements in our intervention program. In order to increase the likelihood of an effective peer-learning method, we further tailored every element to the target group, for example, by adjusting the vocabulary, glossary, and *S. suis* case studies in the e-

learning module. This would not have been possible without the involvement of practicing swine veterinarians and an experienced Implementation Mapping project team throughout the whole development process.

A possible limitation of our intervention program is the way in which we incorporated the change objectives in our program without prioritizing them. Although determining the importance and changeability of the change objectives is part of Implementation Mapping [20], we were not able to do this because of the lack of evidence-based information. Spending more time on the change objectives that are difficult to change (e.g., scoring how many times the change objectives are present in the program using different theoretical methods) could increase the effectiveness of the intervention program.

Implementation Mapping helped to address the changeable objectives in our program, but there are also objectives that are not (easily) changeable. Antimicrobial stewardship is a complex multifaceted health issue influenced by interactions between many different factors and actors, e.g., the consumers who want to purchase animal products at a low cost, the meat processing industry, the supermarkets, and the farmers who are producing under financial pressure [45,46]. It is important to point out that the *S. suis* guideline was developed for and with swine veterinarians in practice [11] and that the scope of our intervention program was to change individual veterinarians' behavior towards better implementation of the guideline. Individual veterinarians are indeed key stakeholders, but they do not always control other stakeholders' behavior. Veterinarians' uptake of antimicrobial stewardship principles can also be influenced by the opinions and behaviors of farmers and other farm advisors [47]. Although farmers and other farm advisors were not addressed directly in our intervention, we did identify change objectives describing how veterinarians can deal with them indirectly, as the veterinarian appears to be the most important source of information for sow farmers [48]. For example, veterinarians can be trained on how to deal with a farmer's resistance to accepting and following their advice or how to maintain good relations with a farmer's other advisors. This was translated in the program as peer feedback to ask other veterinarians for tips and tricks for their *S. suis* problem farms.

In a similar vein, (inter)national rules and regulations play an important role in antimicrobial stewardship in general, as well as in adherence to clinical guidelines. For instance, our findings, along with those of others [49], indicate that some practicing swine veterinarians had negative experiences with rules and regulations—upon which in their eyes the veterinary guidelines were built—regarding antimicrobial resistance. Although rules and regulations can be relevant barriers to implementing antimicrobial stewardship principles [26] (e.g., in relation to building or renovating pig pens, which can contribute to better biosecurity or housing conditions but normally is time consuming because of all kinds of procedures), changing these instruments is a complex policy and political endeavor that is beyond the influence of individual swine veterinarians. Therefore, rules and regulations were not included in the change and performance objectives of this intervention program. In summary, combatting against antimicrobial resistance is complex, and many interactive factors and actors play a role [50]. The veterinarian has a key role but also functions as a cog in a much bigger machinery.

The same dependence on others plays a role in the maintenance (e.g., evaluation, updating) of the veterinary guidelines in the Netherlands. We know that one of the core principles regarding good clinical practice guidelines is to continually revise them to keep them updated to the current situation [51]. However, the veterinary profession in the Netherlands has not yet succeeded in developing a (financial) plan for the maintenance of the veterinary clinical practice guidelines; this is a familiar problem in human medicine also [19]. The maintenance of veterinary clinical practice guidelines is a big challenge [52] and could reduce the effect of our intervention program, as changing the quality of the guideline is difficult, and the development

of a sustainable plan for the future of our intervention program is even harder to achieve. A solution could be to incorporate the veterinary guidelines in a larger system of veterinary quality control (e.g., an independent institute [52]).

It should furthermore be noted that, despite the positive sides of the Implementation Mapping approach, a downside is its time-intensive nature and the need to involve a large number of individuals, which is not always financially or practically possible and could result in a loss of momentum. The development process of our intervention program took 18 months and involved a project team, multiple adopters, and stakeholders. Intervention Mapping is a time-consuming process—reflecting also the difficulty of changing behaviors that cannot be resolved without a comprehensive process—but it brings the development of interventions to a higher level [53]. We believe that this process is necessary for the *S. suis* guideline because of its implementation gap (research-to-practice gap).

If there is no (financial and/or practical) possibility of following the Implementation Mapping approach, a shorter process (e.g., rapid Implementation Mapping [54]) could be effective enough to improve or maintain adherence to the guideline. To our knowledge, this is the first time that a theory-based intervention program for a veterinary guideline has been developed, and our results and experiences could serve as a protocol for designing interventions to support adherence to diverse guidelines in animal health promotion. For example, our change and performance objectives could be used as an example to speed up the process, and some of our results can shorten the process (e.g., experience gained with the process, knowledge about possible beliefs of adopters, and program activities). However, it is important to point out that tasks cannot be completely skipped without the risk of missing aspects that could decrease the effectiveness of the program.

Overall, the use of Implementation Mapping has served as a useful framework to ensure the integration of theory, evidence, and existing practice in the veterinary sector to develop a behavior change intervention.

## Conclusion

This paper demonstrates that the Implementation Mapping approach supported the systematic development of a theory-based intervention program to increase veterinarians' adherence to the *S. suis* guideline for weaned pigs. Our intervention program includes peer-learning meetings with self-evaluation and feedback based on performance indicators, an e-learning module, case discussions, a quiz, and movies from a practicing swine veterinarian and specialists. Because there were financial and time limits in this study, we had to make practical choices. Implementation Mapping is a comprehensive systematic approach, and, when followed step-by step in an iterative way, it serves to ensure that relevant and clear objectives are set, that relevant and changeable behavioral determinants for reaching the objectives are identified, and that these are properly addressed through theory- and evidence-based methods in the intervention program.

## Supporting information

**S1 Fig. Print screen e-learning *S. suis* guideline.** Example method active learning.
(TIF)

**S2 Fig. Print screen e-learning *S. suis* guideline.** Example method persuasive communication.
(TIF)

**S1 File. Questions knowledge quiz.** S1 Table. Matrix of performance objectives linked to the change objectives.
(DOCX)

## Acknowledgments

We are very grateful to the participating veterinarians, farmers, and other stakeholders for their time. We also want to thank the Dutch Institute for the Rational Use of Medicine for its cooperation.

## Author Contributions

**Conceptualization:** Isaura Y. A. Wayop, Emely de Vet, Joanne N. Leerlooijer, Jaap A. Wagenaar, David C. Speksnijder.

**Data curation:** Isaura Y. A. Wayop, Emely de Vet, Jaap A. Wagenaar, David C. Speksnijder.

**Formal analysis:** Isaura Y. A. Wayop, Emely de Vet.

**Funding acquisition:** Isaura Y. A. Wayop, Emely de Vet, Jaap A. Wagenaar, David C. Speksnijder.

**Investigation:** Isaura Y. A. Wayop.

**Methodology:** Isaura Y. A. Wayop, Emely de Vet, Joanne N. Leerlooijer.

**Project administration:** Isaura Y. A. Wayop, Jaap A. Wagenaar, David C. Speksnijder.

**Resources:** Isaura Y. A. Wayop.

**Software:** Isaura Y. A. Wayop.

**Supervision:** Emely de Vet, Jaap A. Wagenaar, David C. Speksnijder.

**Validation:** Isaura Y. A. Wayop, Emely de Vet, Jaap A. Wagenaar, David C. Speksnijder.

**Visualization:** Isaura Y. A. Wayop.

**Writing – original draft:** Isaura Y. A. Wayop.

**Writing – review & editing:** Isaura Y. A. Wayop, Emely de Vet, Joanne N. Leerlooijer, Jaap A. Wagenaar, David C. Speksnijder.

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
