## [Decision Letter · Decision Letter 0]

15 Nov 2023

PONE-D-23-32153Using Implementation Mapping to develop an intervention program to support veterinarians’ adherence to the guideline on *Streptococcus suis* clinical practice in weaned pigsPLOS ONE

Dear Dr. Speksnijder,

Thank you for submitting your manuscript to PLOS ONE. After careful consideration, we feel that it has merit but does not fully meet PLOS ONE’s publication criteria as it currently stands. Therefore, we invite you to submit a revised version of the manuscript that addresses the points raised during the review process.

We look forward to receiving your revised manuscript.

Kind regards,

Gianmarco Ferrara, PhD, MVD

Academic Editor

PLOS ONE

Journal Requirements:

"The VET-ENHANCE project (Veterinary guidelines to support antimicrobial stewardship: enhancing implementation through behavioural interventions) was funded by ZonMw (The Netherlands Organisation for Health Research and Development), project number: 541002005."

3. We notice that your supplementary figures are uploaded with the file type 'Figure'. Please amend the file type to 'Supporting Information'. Please ensure that each Supporting Information file has a legend listed in the manuscript after the references list.

**Additional Editor Comments:**

Both reviewers complained about a lack of specificity and concreteness in these guidelines. The authors are requested to improve the manuscript by following the comments listed by the reviewers.

Reviewers' comments:

Reviewer's Responses to Questions

**Comments to the Author**

1. Is the manuscript technically sound, and do the data support the conclusions?

Reviewer #1: Partly

Reviewer #2: Yes

2. Has the statistical analysis been performed appropriately and rigorously? 

Reviewer #1: N/A

Reviewer #2: N/A

3. Have the authors made all data underlying the findings in their manuscript fully available?

Reviewer #1: Yes

Reviewer #2: Yes

4. Is the manuscript presented in an intelligible fashion and written in standard English?

Reviewer #1: Yes

Reviewer #2: Yes

5. Review Comments to the Author

Reviewer #1: This is to me an atypical manuscript, since it is more a guide than a scientific paper.

I do not have major comments. I would say that S. suis here seems to be an excuse since most recommendations may be applied to most bacterial infectious diseases...and not specifically for S. suis.

Lines 76-83: I can understand that the previous guide was not followed...and I am not sure this one will be...but it is a good idea to try...

Lines 137-143: it is strange that no expert on S. suis infections has been contacted...it is a highly complicated infection, with no universal markers to identify virulent strains and interventions will be different if a virulent strain is present or not...S. suis may be a primary or a secondary agent.

Lines 147-150: this may be true for infections other than S. suis...

Table 2, 1b: the final diagnostic of a true S. suis infection must be done through the lab...so the word "possible" here is not correct...it should be "mandatory". It is highly necessariy to perform bacterial isolation and serotyping, at minimum.

Table 2, 7: the number of tested piglets should be higher and repeated at least three times in a relatively short period of time: it is not easy to diagnose a S. suis problem and sometimes different serotypes are found in the same herd at the same time, which is an indication of S. suis not being a primary agent...actions to be taken will be different from a herd with the presence of a highly virulent strain...

Table 2, 10: why not arthritis?

Reviewer #2: General comments

The manuscript is well-written and very comprehensive towards the process followed to create and distribute the message on a guideline for Streptococcus suis clinical practice in pigs. Although I can follow the reasoning very closely, I end the review of this manuscript with a lack of ‘what was the final results of these efforts now?’. What I mean is that following the very thorough description of the process gone through to develop the intervention program, the authors state the started implementing the specific groups at the end of the paper, but they do not report any outcome of this large effort made towards the ‘real’ adherence that resulted from the roll-out of the program. Furthermore, I have some more specific comments that need to be addressed.

Specific comments

L83 Please adapt ‘at farms’ by ‘on farms’.

L161 I would rather say ‘In summary, …’ which is more commonly used to come up with a conclusions.

L192 Point 9 of the table. I would prefer to use ‘euthanasia’ instead of the currently used ‘euthanization’ which sounds very unfamiliar to me.

L195 I would exchange ‘holdings’ by ‘farms’.

L392 I presume that ‘imported’ should be ‘important’.

L394 Please correct ‘towards’.

L405 I never heard of the statement ‘In a similar vein, …’. I presume this is a kind of mistake (autosuggestion tool in WORD) and should rather be ‘In a similar way, …’

L406 The sentence ‘For example, our and other results …’ sound a bit weird. Please rephrase in a better readable way.

L410 I have never heard of ‘pigsties’. I presume you mean ‘post-weaning facilities’ which should be a word that all interested readers can understand.

L414 Please write ‘In summary, …’.

L414 The phrase ‘combatting the fight against …’ sound as a double statement. I would suggest to come up with something like ‘In summary, combatting against …’.

L416 Please correct the sentence to ‘The veterinarian has a key role but also functions a a cog in a much bigger machinery.’

L419 Please rephrase ‘… updated and current’ by ‘… updated to the current situation’.

L421 Move ‘also’ from the end of the sentence to ‘… this is also a familiar problem …’.

6. PLOS authors have the option to publish the peer review history of their article (what does this mean?). If published, this will include your full peer review and any attached files.

Reviewer #1: No

Reviewer #2: No

---

## [Author Response · Author response to Decision Letter 0]

16 Jan 2024

Response to Reviewers

PONE-D-23-32153

Using Implementation Mapping to develop an intervention program to support veterinarians’ adherence to the guideline on Streptococcus suis clinical practice in weaned pigs

PLOS ONE

Both reviewers complained about a lack of specificity and concreteness in these guidelines. The authors are requested to improve the manuscript by following the comments listed by the reviewers.

We thank the reviewers for their efforts to review our manuscript and providing valuable feedback for improvements. We have thoroughly read the feedback and recommendations and adapted the manuscript accordingly. We respond to the reviewers’ comments point by point below.

Comments to the Author

1. Is the manuscript technically sound, and do the data support the conclusions?

Reviewer #1: Partly

Reviewer #2: Yes

2. Has the statistical analysis been performed appropriately and rigorously? 

Reviewer #1: N/A

Reviewer #2: N/A

3. Have the authors made all data underlying the findings in their manuscript fully available?

Reviewer #1: Yes

Reviewer #2: Yes

4. Is the manuscript presented in an intelligible fashion and written in standard English?

Reviewer #1: Yes

Reviewer #2: Yes

5. Review Comments to the Author

Reviewer #1: This is to me an atypical manuscript, since it is more a guide than a scientific paper. I do not have major comments. I would say that S. suis here seems to be an excuse since most recommendations may be applied to most bacterial infectious diseases...and not specifically for S. suis.

We thank Reviewer 1 for his/her efforts to review our manuscript and providing feedback for improvements.

This is the first time Implementation Mapping has been used in veterinary medicine to support the implementation of clinical practice guidelines. We used the existing clinical guideline of S. suis as the point of departure for illustrating how Implementation Mapping can be used to develop an intervention to stimulate the implementation of a clinical guideline in practice. While clinical guidelines are widely recognized as valuable components of antimicrobial stewardship programs in veterinary medicine worldwide, a gap exists in understanding how these guidelines can be effectively implemented in veterinary practice. Our aim is to address and fill this knowledge gap which is made more explicit in our manuscript, line 96-99. 

Similar papers, in the domain of human medical practice have been published before in PLOS ONE (e.g.: BREATHLEssness in INDIA (BREATHE-INDIA)–Study protocol for the co-design of a community breathlessness intervention in India using realist methods and intervention mapping https://doi.org/10.1371/journal.pone.0293918).

Importantly, we did not develop the clinical guideline for S. suis ourselves. This guideline had been published in 2014 by The Royal Dutch Society of Veterinary Medicine (KNMvD). However, the regular use of the guideline in practice is limited and we aimed to develop an accompanying intervention to support adherence to the veterinary guidelines in practice. The distinction between the existing S. suis guideline and the development of an implementation intervention to support adherence to the guideline is made more explicit in our manuscript, line 20-22, 26-31 and 96-99. 

We agree that most recommendations in the S. suis guideline can be applied for most bacterial infectious diseases. Therefore our results are relevant to a broader scope of infections than S. suis alone and the intervention might be used for other infectious diseases. However, we do know that S. suis infections are seen as one of the major drivers of antimicrobial use in the pig sector, and the antimicrobial resistance rates in S. suis have increased worldwide. We describe this also in our introduction in the manuscript, line 70-79. 

Lines 76-83: I can understand that the previous guide was not followed...and I am not sure this one will be...but it is a good idea to try...

You are right that the S. suis guideline was used only partly or not at all in practice. We did not make any changes to the guideline itself or develop a new one ourselves, as that is the responsibility of the guideline developer when new scientific or other evidence arises. To make this more clear we made a change, see line 20-22, 26-31 and 96-99. Our aim was to improve adherence to the existing clinical guideline by developing an intervention program to support veterinarians in following the existing guideline. This approach has proved its succes in human medicine, line 107-110.

Lines 137-143: it is strange that no expert on S. suis infections has been contacted...it is a highly complicated infection, with no universal markers to identify virulent strains and interventions will be different if a virulent strain is present or not...S. suis may be a primary or a secondary agent.

We fully agree with this statement about the complexity. The S. suis guideline is indeed developed with experts on S. suis infections from 2012-2014. We made this more explicit, see line 144

Lines 147-150: this may be true for infections other than S. suis...

We do agree that this example can also be used for other infections, but our aim in this study was to critically study barriers for implementation of the S. suis guideline. Our results showed that this example is also the case for S. suis as farmers can have difficulties with giving individual treatments (see also the article: Why Veterinarians (Do Not) Adhere to the Clinical Practice Streptococcus suis in Weaned Pigs Guideline: A Qualitative Study https://doi.org/10.3390/antibiotics12020320). In the introduction we added that the statements in the S. suis guideline are indeed applicable to other infections, lines 81-82. 

Table 2, 1b: the final diagnostic of a true S. suis infection must be done through the lab...so the word "possible" here is not correct...it should be "mandatory". It is highly necessariy to perform bacterial isolation and serotyping, at minimum.

The information here describes our performance objective based on what is written in the S. suis guideline, see also our explanation at the first point you raised. Here we want to point out that if bacterial examination or other diagnostics are performed this should be reported. To make this clear we changed the word possible to potential. Table 2.1b Line 198 and in the Supporting Information, S1.1b

Table 2, 7: the number of tested piglets should be higher and repeated at least three times in a relatively short period of time: it is not easy to diagnose a S. suis problem and sometimes different serotypes are found in the same herd at the same time, which is an indication of S. suis not being a primary agent...actions to be taken will be different from a herd with the presence of a highly virulent strain…

Thank you for this recommendation. However, we have not made changes to statements of the S. suis guideline itself. Please also see our earlier explanation that the guideline was already established prior to our work based on a rigorous process involving swine veterinarians and S. suis experts. The focus of our paper is on the development of a strategy to accompany the implementation of the guideline in order to support its adherence.

Table 2, 10: why not arthritis?

The S. suis guideline in the Netherlands is only written for meningitis because of the different treatment plan. This is however outside the scope of our research as we are focused on this existing S. suis guideline, please see also our previous answers. 

Reviewer #2: General comments

The manuscript is well-written and very comprehensive towards the process followed to create and distribute the message on a guideline for Streptococcus suis clinical practice in pigs. Although I can follow the reasoning very closely, I end the review of this manuscript with a lack of ‘what was the final results of these efforts now?’. What I mean is that following the very thorough description of the process gone through to develop the intervention program, the authors state the started implementing the specific groups at the end of the paper, but they do not report any outcome of this large effort made towards the ‘real’ adherence that resulted from the roll-out of the program. 

We thank Reviewer 2 for his/her efforts to review our manuscript and positive feedback about the reasoning of the process. The final outcome of this article is the intervention program. We do not have the full results of the evaluation of the intervention program trial yet. We aimed to describe the implementation mapping process leading to our outcome, the intervention program. We made this aim more explicit in the introduction, line 96-99. Also to present more clear results, we have added more insights into the actual elements of the intervention program in the article itself by adjusting Table 6, line 309-313. 

Furthermore, I have some more specific comments that need to be addressed.

Specific comments

L83 Please adapt ‘at farms’ by ‘on farms’.

We have done so accordingly. Line 86

L161 I would rather say ‘In summary, …’ which is more commonly used to come up with a conclusions.

We have adapted the paper accordingly. Line 167

L192 Point 9 of the table. I would prefer to use ‘euthanasia’ instead of the currently used ‘euthanization’ which sounds very unfamiliar to me.

We have done so accordingly. Line 198 Table 2.9

L195 I would exchange ‘holdings’ by ‘farms’.

We have done so accordingly. Line 200-201

L392 I presume that ‘imported’ should be ‘important’.

We have done accordingly. Line 402

L394 Please correct ‘towards’.

We have done accordingly. Line 404

L405 I never heard of the statement ‘In a similar vein, …’. I presume this is a kind of mistake (autosuggestion tool in WORD) and should rather be ‘In a similar way, …’

"In a similar vein" and "In the same way" convey a similar idea but have slightly different nuances.

"In a similar vein" suggests a similarity or connection in terms of the overall theme or approach, but it doesn't necessarily imply an identical method or process. "In the same way" is more focused on expressing a similarity in the manner or method of doing something. It emphasizes the similarity in the approach or procedure. (https://idioms.thefreedictionary.com/in+a+similar+vein)

We choose to use “in a similar vein”, in consultation with an English native speaker, as this is the nuance we want to give in this sentence. 

L406 The sentence ‘For example, our and other results …’ sound a bit weird. Please rephrase in a better readable way.

We have rephrased the sentence. Line 416-417

L410 I have never heard of ‘pigsties’. I presume you mean ‘post-weaning facilities’ which should be a word that all interested readers can understand.

We mean all kind of pig pens and adjusted this. Line 420

L414 Please write ‘In summary, …’.

We have done accordingly. Line 424

L414 The phrase ‘combatting the fight against …’ sound as a double statement. I would suggest to come up with something like ‘In summary, combatting against …’.

We have done accordingly. Line 425

L416 Please correct the sentence to ‘The veterinarian has a key role but also functions a a cog in a much bigger machinery.’

We have done accordingly. Line 426-427

L419 Please rephrase ‘… updated and current’ by ‘… updated to the current situation’.

We have done accordingly. Line 429-430

L421 Move ‘also’ from the end of the sentence to ‘… this is also a familiar problem …’.

Thank you for this recommendation. We have decided to leave the sentence as it is due to its focus on human medicine, as we intended."

---

## [Editor Report · Decision Letter 1]

19 Feb 2024

Using Implementation Mapping to develop an intervention program to support veterinarians’ adherence to the guideline on *Streptococcus suis* clinical practice in weaned pigs

PONE-D-23-32153R1

Dear Dr. Speksnijder,

We’re pleased to inform you that your manuscript has been judged scientifically suitable for publication and will be formally accepted for publication once it meets all outstanding technical requirements.

Kind regards,

Gianmarco Ferrara, PhD, MVD

Academic Editor

PLOS ONE
---

## [Editor Report · Acceptance letter]

23 Feb 2024

PONE-D-23-32153R1 

PLOS ONE

Dear Dr. Speksnijder, 

I'm pleased to inform you that your manuscript has been deemed suitable for publication in PLOS ONE. Congratulations! Your manuscript is now being handed over to our production team.

Kind regards, 

on behalf of

Dr. Gianmarco Ferrara 

Academic Editor

PLOS ONE